# Development of diagnostic SNP markers for quality assurance and control in sweetpotato [*Ipomoea batatas* (L.) Lam.] breeding programs

Dorcus C. Gemenet[1]*, Mercy N. Kitavi[1], Maria David[2], Dorcah Ndege[1], Reuben T. Ssali[3], Jolien Swanckaert[4], Godwill Makunde[5], G. Craig Yencho[6], Wolfgang Gruneberg[2], Edward Carey[3], Robert O. Mwanga[4], Maria I. Andrade[5], Simon Heck[1], Hugo Campos[2]

**1** International Potato Center (CIP), ILRI Campus, Nairobi, Kenya, **2** International Potato Center (CIP), Apartado, Lima, Peru, **3** International Potato Center (CIP), Kumasi, Ghana, **4** International Potato Center (CIP), Kampala, Uganda, **5** International Potato Center (CIP), Maputo, Mozambique, **6** North Carolina State University, Raleigh, North Carolina, United States of America

\* d.gemenet@cgiar.org

**Data Availability Statement:** The data underlying the results presented in the study are provided as

## Abstract

Quality assurance and control (QA/QC) is an essential element of a breeding program's optimization efforts towards increased genetic gains. Due to auto-hexaploid genome complexity, a low-cost marker platform for routine QA/QC in sweetpotato breeding programs is still unavailable. We used 662 parents of the International Potato Center (CIP)'s global breeding program spanning Peru, Uganda, Mozambique and Ghana, to develop a low-density highly informative single nucleotide polymorphism (SNP) marker set to be deployed for routine QA/QC. Segregation of the selected 30 SNPs (two SNPs per base chromosome) in a recombined breeding population was evaluated using 282 progeny from some of the parents above. The progeny were replicated from *in-vitro*, screenhouse and field, and the selected SNP-set was confirmed to identify relatively similar mislabeling error rates as a high density SNP-set of 10,159 markers. Six additional trait-specific markers were added to the selected SNP set from previous quantitative trait loci mapping studies. The 36-SNP set will be deployed for QA/QC in breeding pipelines and in fingerprinting of advanced clones or released varieties to monitor genetic gains in famers' fields. The study also enabled evaluation of CIP's global breeding population structure and the effect of some of the most devastating stresses like sweetpotato virus disease on genetic variation management. These results will inform future deployment of genomic selection in sweetpotato.

## Introduction

Development of user-friendly, low-cost, high-throughput markers for quality assurance and control (QA/QC) in a genomic-assisted breeding (GAB) era is a critically important aspect in

supplementary information together with the submission.

**Funding:** Genotyping for this work was funded by the SUSTAIN project awarded to the International Potato Center (CIP) (SH) by Department for International Development (DFID) through grant Number (1198-DFID). Most co-authors were supported by a Bill & Melinda Gates Foundation (BMGF) grant (Grant number OPP1052983) awarded to North Carolina State University (GCY). The funders had no role in study design, data collection and analysis, decision to publish, or preparation of the manuscript.

**Competing interests:** The authors have declared that no competing interests exist.

crop improvement and germplasm conservation [1,2]. This is because genetic fidelity and trueness-to-type are often not phenotypically obvious. The use of molecular markers for QA/QC has been implemented in several plants and animals and single nucleotide polymorphism (SNP) markers have become the markers of choice in germplasm characterization and QA/QC [3]. For example, [4] developed transcriptome-derived SNP markers for cost-efficient forest seed stock identification, [5] developed 'near-minimal' sets of SNPs to differentiate operational taxonomic units in fruit flies, while [6] developed species-diagnostic SNP markers to analyze admixture structure of varieties and rootstocks in citrus. Here we define QA and QC according to [7] who defined QA as the process or set of processes used to measure the quality of a product and QC as the process of ensuring products and services meet consumer expectations. In plant breeding, QA would refer to all measures put in place to prevent errors and create a high-quality variety, while QC is the process of identifying the defects or errors in the quality of the breeding line, germplasm accession, variety, or any other product from the breeding pipeline. Solid QA/QC procedures are critical in plant breeding, as errors in the process of developing new varieties can lead to wasted time and resources, and also reduce and/or cancel out genetic gains achieved because of genotype mix-ups along the breeding pipeline.

For most crops, the same SNP set can be used in a QA program to characterize germplasm, and study genetic diversity, genetic relationships and population structure, and in a QC program to evaluate genetic identity, genetic purity, parent-offspring identity, validation of crosses in nurseries and trait-specific testing [3, 7]. Genotype misclassification is a common problem in most crops as has been reported in *Oryza spp* [8] and *Brassica spp* [9], that has consequences in breeding and variety development. QA/QC have become even more important with the advent of molecular markers for decision support in breeding programs. Whereas the importance of QC filtration methods for SNP markers are well established [10,11] and methods put in place, QA/QC of the phenotypes that are combined with genotypes to predict performance are generally not very well established. The general lack of phenotype/genotype concordance has led to molecular decision support tools for increasing genetic gains in plant breeding such as genomic selection not achieving their full potential. Although well acknowledged in human and animal genetic fields [12,13], reports on the effects of poor QA/QC and genotype misclassification in plant breeding are generally lacking. With next- and third-generation sequencing methods enhancing rapid and cost-efficient development of large amounts of genomic data [11, 14], one of the biggest challenges of plant breeding programs is putting in place highly precise mechanisms for QA/QC of the phenotyping/genotyping processes.

Sweetpotato is a crop of increasing importance in sub-Saharan Africa (SSA) contributing to both food and nutritional security, from both adapted, starchy, white-fleshed varieties and new improved high β-carotene, orange-fleshed varieties [15–17]. The International Potato Center (CIP), one of the centers of the Consultative Group on International Agricultural Research (CGIAR), runs a global sweetpotato improvement program. CIP is headquartered in Lima, Peru and has established three additional breeding support platforms in SSA. The support platform at Lima offers global technical support, while the east and central Africa platform focusses on end-user preferred varieties within this region including resistance to sweetpotato virus disease (SPVD), a major production constraint within the region. The southern Africa breeding support platform focusses on end-user preferred varieties in addition to drought tolerance, which is the major production constraint in this region, while the west African support platform focusses on culinary aspects, especially the 'less sweet' sweetpotato which is preferred in this region [17]. Being mainly an auto-hexaploid, GAB tools are just starting to be mainstreamed into the breeding program due to genome complexity. Several genomic tools have been developed, in partnership with several development partners and a molecular breeding team is currently stationed at CIP's regional office for SSA in Nairobi, Kenya to facilitate this

mainstreaming. Putting in place QA/QC measures is prerequisite to enhance the likelihood of success applying GAB in sweetpotato breeding. The main objectives of the current study were: i) to characterize breeding population parents from global support platforms for population structure, ii) to estimate allele diversity among the breeding population parents from the four global support platforms iii) to estimate levels of misclassification error through different steps of a multi-stage, multi-country breeding pipeline, iv) to develop a low-cost diagnostic SNP set for rapid QA/QC of sweetpotato breeding populations.

## Materials and methods

### Genetic materials

We collected parents from all four global breeding support platforms of CIP: Peru, being the global support platform; Uganda, being the support platform for east and central Africa; Mozambique, being the support platform for southern Africa; and Ghana, being the support platform for west Africa. We had 331 parents from Peru, 126 parents from Uganda, 144 parents from Mozambique and 61 parents from Ghana, totaling 662 parents. The list of the breeding population parents is provided in **S1 Table**. Since our objective was to mainstream QA/QC in breeding trials, we used progeny from a segregating breeding population to validate that the finally selected SNP set segregates in recombined individuals from parents and can identify relatively the same error rates as a high-density SNP set. These validation materials were derived from a breeding population progeny developed from the east and central African support platform, named the Mwanga Diversity Panel (MDP).

### Genotyping and SNP calling

DNA from the breeding population parents and segregating progeny was extracted at the Biosciences east and central Africa (BecA) laboratories based at the International Livestock Research Institute (ILRI), Nairobi. The extraction was done following a modified cetyl tri-methylammonium bromide (CTAB) method optimized for sweetpotato. The DNA was treated for contaminating RNA using RNAse A, quantified and normalized using standard protocols. The DNA was then submitted for sequencing using the Diversity Array Technology's DArT-Seq method implemented by BecA's Integrated Genotyping Service and Support platform (IGSS) as described by [18], as separate genotyping projects. IGSS is a subsidized genotyping platform supported by the Bill and Melinda Gates Foundation to enhance use of genomics in breeding for SSA. Sequencing was done at 96-plex, high density and SNP calling done using DArT's proprietary software DArTSoft [18], with aligning to the diploid reference genome of *Ipomoea trifida*, a relative of sweetpotato [19]. Given that most commercial genotyping platforms have allele depth coverage ~25x to 30x, previous studies [20] have shown that this depth of coverage is not adequate to call allele dosage with confidence in genotype quality for hexaploid sweetpotato. The study also showed that in such cases, 'diploidized' biallelic loci which are informative enough performed almost as well as data with high confidence dosage information, for simple traits. Therefore, biallelic markers used in this study were called in a diploidized version which means that the five heterozygous classes expected in sweetpotato were collapsed into one heterozygous class and the SNPs were coded as 0 = AA, 1 = BB, 2 = AB and "-"= Missing. All genotype-quality related parameters including call rate, frequency of loci homozygous for the reference allele (FreqHomRef), frequency of loci homozygous for the SNP allele (FreqHomSnp), frequency of heterozygous loci (FreqHets), polymorphic information content of the SNP allele (PICSnp) and average polymorphic information content (AvgPIC) were computed as part of the DArT's proprietary software DArTSoft workflow. The metadata

defining data columns as obtained from the DArTseq platform is provided as **S1 File**. The raw data from the global parents is provided as **S1 Data**.

## Data analysis and validation

**Population structure of International Potato Center's breeding parents.** Since allele frequencies through genotype calling are biased when allele depth of coverage is relatively low [21] and given that we used diploidized markers for a hexaploid, we used non-parametric methods as described by [22], to estimate allele sharing distance (ASD). These methods do not assume Hardy-Weinberg equilibrium or linkage equilibrium and were implemented using the program AWClust 3.1. The phylogenetic tree was constructed using MEGA X program [23]. For allele diversity, Nei's coefficients of inbreeding ($F_{IS}$) [24] and Wright's inbreeding coefficients ($F_{ST}$ or θ) according to [25], were estimated in R. Additionally, linkage disequilibrium (LD) between pairs of markers used for parental population structure was done using the LDheatmap package in R, with the option of estimating $r^2$. The LD between pairs of markers was carried out to evaluate the distribution and representativeness in the genome of the SNPs used in structure and diversity analysis, rather than to estimate the LD decay in sweetpotato, which would require a higher-density of SNPs and other considerations for allele dosage, linkage phases and haplotypes.

**Estimating the level of misclassification error during breeding operations and field experiments.** We used the MDP multi-family breeding population to estimate the level of error expected from our standard cross-border breeding operations. This population was developed by crossing 8B by 8A parents without reciprocals, from the east and central Africa breeding platform in CIP-Uganda, thereby resulting in 64 families, with about 30 progeny per family on average. The crossing was done following a $B^*A$ pseudo-heterotic grouping based on genetic distance established by simple sequence repeat (SSR) markers [26]. The seed was germinated and maintained *in vitro* at the Biosciences east and central Africa (BecA) hub by CIP-Kenya, from where the *in vitro* plantlets were shipped back to CIP-Uganda for screenhouse multiplication and field experiments at Namulonge (0˚ 31' 17.99" N, 32˚ 36' 32.39" E). To evaluate the genetic fidelity following movement from *in vitro* to screenhouse and then to field experiments, we randomly sampled 94 genotypes from 1,886 genotypes of the MDP, which constitutes about 5% of the population, and genotyped using DArTSeq, as described above. The genotyping was done in 96-plex resulting in three 96-well plates (one each for *in vitro*, screen house and field) and a total of 282 DNA samples (**S2 Table**). The raw data from the segregating progeny is provided as **S2 Data**. The SNPs were filtered and used in further analysis. The ASD was computed as for the parents above [22] and a phylogenetic tree was generated using DARwin 6.0.21 [27]. Afterwards the clustering was examined based on positions on the tree and Sankey diagrams developed using the Alluvial package [28] in R. Additionally, Monte-Carlo tests were used to calculate the correlation among the separate distance matrices from *in vitro*, screenhouse and field using the mantel.rtest function of ade4 package in R.

**Selection and validation of a diagnostic SNP set.** For our objective to develop a QA/QC SNP set that would be diagnostic for the global breeding population, we selected SNPs identified from parents as described above but also validated the selected SNPs using segregating progeny from a breeding population. We therefore selected only high-quality SNP markers that were present in both the parents and the MDP breeding population progeny. The selected markers were confirmed if they kept the same population structure of the parents and still identified the same misclassification error rate in the MDP population. Several studies have selected rapid QA/QC sets with as low as 10 SNP markers e.g. in Maize [29]. However, the

base chromosome number of sweetpotato is 15 and given that we were diploidizing hexaploid loci, our aim was to have a minimum of two markers per base chromosome. We performed principal component analysis of the intermediate marker set according to [29], but no apparent grouping of the markers was determined. We therefore selected the final 30 SNP markers based on chromosome number and genetic distance, from an intermediate marker set of 85 SNP markers. To establish the utility of the 30 selected SNPs for rapid QA/QC, we compared the ASD of both parents and MDP populations based on the 30 selected SNPs and the ASD based on their respective original filtered marker sets (205 SNPs for parents and 10,159 SNPs for MDP), using DARwin 6.0.21 tree comparison function [27]. The filtered data, including the 205-SNP set, 85-SNP set, and 85-SNP set for MDP are provided in **S3 Data**.

## Results

### SNP profile from the parental and segregating populations

A total of 9,670 SNP markers were obtained from genotyping the 662 parents of CIP's breeding population (**S1 Data**). The unfiltered dataset had call rates ranging from 0.4–1.0, FreqHomRef ranging from 0.0–1.0, FreqHomSnp ranging from 0.0–1.0, FreqHets ranging from 0.0–0.3, PICSnp and AvgPIC both ranging between 0.0–0.5 (**Fig 1**). With filtration of $\leq$ 25% missingness, $\geq$ 0.25 polymorphic information content (PIC) and $\geq$ 10% minimum allele frequency (MAF) and an average of 30x allele depth of coverage we recovered 205 SNP markers that were deemed appropriate for analysis of the breeding population structure (**S3 Data**). **Fig 1** shows quality attributes of the unfiltered and filtered SNP data. After filtration, call rates ranged from 0.75–1.0, FreqHomRef ranged from 0.6–0.85, FreqHomSnp ranged from 0.0–0.1, FreqHets ranged from 0.0–0.35, PICSnp ranged from 0.25–0.45, while AvgPIC ranged from 0.1–0.3 (**Fig 1**). The number of filtered SNPs ranged from six to 18 per base chromosome.

The high-density genotyping of the MDP segregating progeny resulted in about 41,194 SNPs (**S2 Data**). Stringent filtering of these based on polymorphic information content (PIC) $\geq$ 0.25, minimum allele frequency $\geq$ 20% and call rate $\geq$ 90%, removing all SNPs with no chromosome position and those on chromosome zero (Chr00) resulted in 10,159 SNPs that were used in further analyses.

### Population structure of CIP's breeding population

We examined population structure of the parents using 205 SNP markers. As expected from a global breeding program, population structuring indicated evidence of germplasm transfer among the breeding support platforms, although there was also evident local adaptation to each support platform (**Fig 2**). The global support platform in Peru had the highest number of parents in the current study. Clustering showed that there is a group of parents from Peru that are closely related to African breeding parents especially those from Ghana and Mozambique. However, an additional group was only unique to Peru (**Fig 2**). This group can also be seen between the first and second dimensions of a 2D multidimensional scale (**S1 Fig**). The east and central Africa support platform in Uganda had a distinct group of parents but also a small admixed group with Mozambique (**Fig 2**). The Uganda platform did not have a lot of admixtures from Peru. Given that the west African support platform was recently established (ca. 2010), and is the smallest in terms of size, the clustering indicates intake of breeding materials from other breeding support platforms especially from Mozambique and Peru, with minimal transfer to Ghana from Uganda. However, on a higher level, the structure can be generalized into two, with one cluster made up of materials from Peru and Ghana, and the other made up of materials from Uganda, Mozambique and Peru.

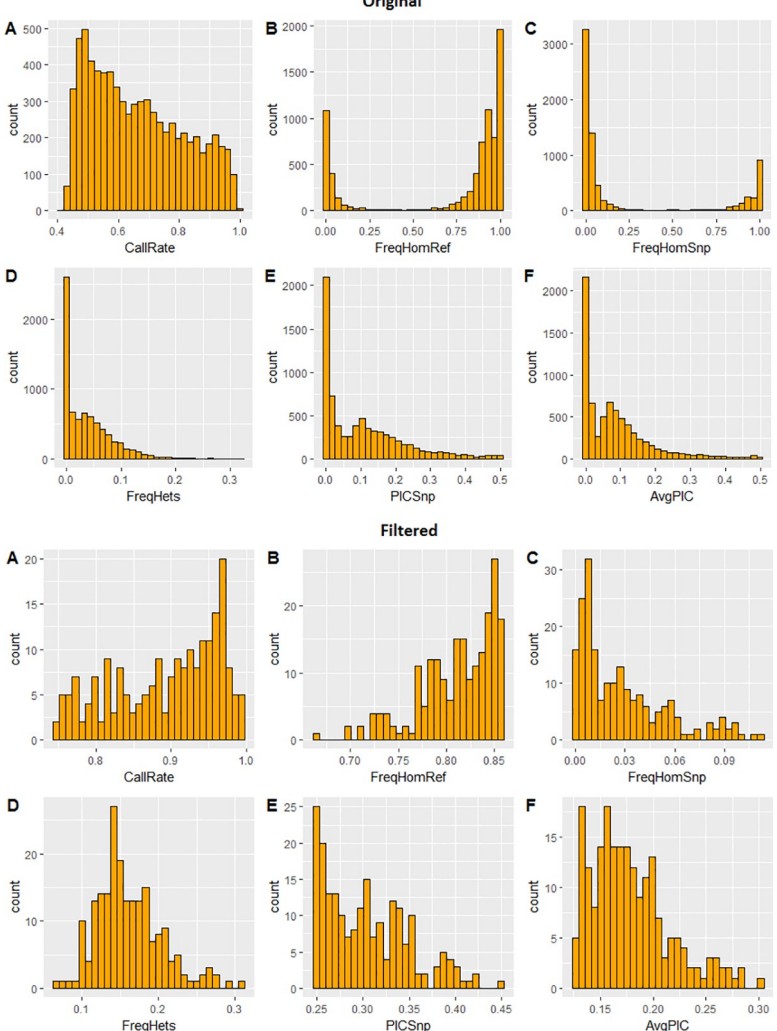

**Fig 1.** Quality attributes of unfiltered (9,670; Top) and filtered (205; Bottom) SNPs from DArTSeq indicating call rate (A), frequency of homozygotes for the reference allele (B), frequency of homozygotes for the alternative allele (SNP; C), frequency of heterozygotes in the data (D), polymorphic information content of the SNP (E), and average polymorphic information content between reference and SNP alleles (F).

## Allele diversity and chromosome-wise linkage disequilibrium between markers

Nei's coefficients of inbreeding using 205 SNPs indicated an average of $F_{IS} = 0.14$ across all populations and that parents from Uganda had the highest inbreeding coefficient $F_{IS} = 0.33$, followed by Mozambique with $F_{IS} = 0.24$. Ghana, followed by Peru had the lowest coefficients of inbreeding at $F_{IS} = 0.008$ and $F_{IS} = 0.07$, respectively. The estimated variance components and fixation indices showed that the correlation of genes within individuals or inbreeding was $F = 0.18$, the correlation of genes in different individuals within the same population was $\theta = 0.07$, and the correlation of genes within individuals within populations $f = 0.12$. Comparing $\theta$ values ($F_{ST}$) between pairs of populations (support platforms in this case) showed that Uganda was the most distinct group with $\theta = 0.08$, $\theta = 0.09$, and $\theta = 0.1$ with Ghana, Mozambique and Peru, respectively. The paired $\theta$ values among Ghana, Mozambique, and Peru were fairly consistent ranging from $\theta = 0.041$ to $\theta = 0.049$. Data is summarized in **Table 1**. We further

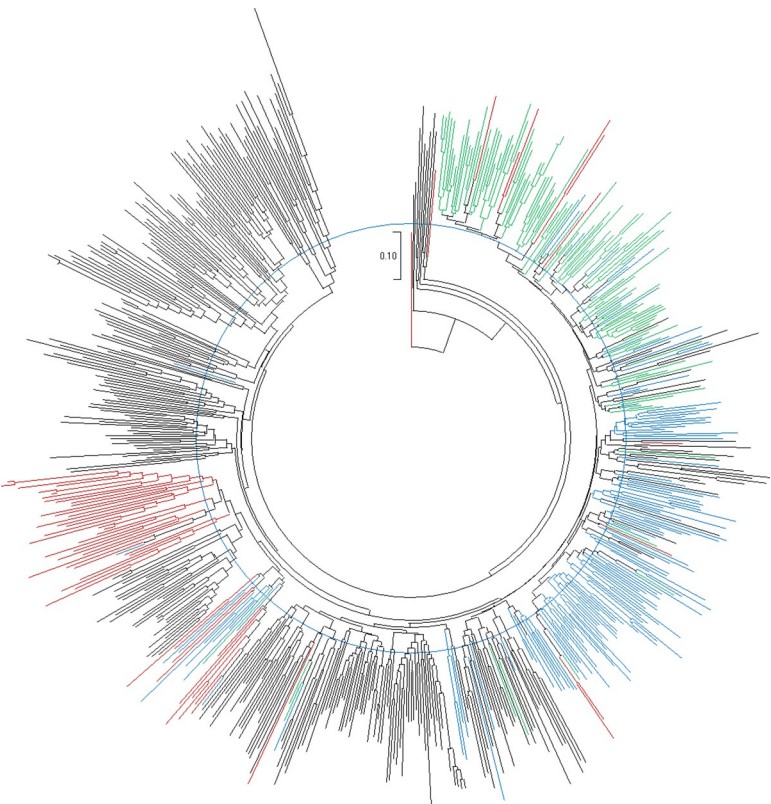

**Fig 2. Phylogenetic tree (Neighbor-Joining) showing the population structure of the International Potato Center (CIP)'s global breeding parents.** Genotypes in Black represent parents from the global support platform in Peru, genotypes in Blue represent parents from the southern Africa support platform in Mozambique, genotypes in Green represent parents from the east and central Africa support platform in Uganda, while genotypes in Red represent parents from the west African support platform in Ghana. The tree was developed using MEGA X.

analyzed for LD between pairs of markers as proxy to check for the distribution of markers genome-wide and per chromosome. Analysis of LD between markers indicated minimal LD among the SNP markers used with the genome-wide LD having an average $r^2 \leq 0.1$. LD per chromosome is presented in **Fig 3.** The results show that very few loci were in LD at $r^2 \geq 0.1$, as majority of loci within a chromosome also had $r^2 \leq 0.1$. These results indicated that the data set was adequately distributed for use in analyzing population structure and diversity. To

**Table 1. Allelic diversity parameters among parents of the International Potato Center (CIP)'s breeding parents from Ghana, Mozambique, Peru and Uganda.**

| Nei's $F_{IS}$ | | | | |
|---|---|---|---|---|
| Ghana | Mozambique | Peru | Uganda | Average |
| 0.008 | 0.24 | 0.07 | 0.33 | 0.14 |
| **Variance and fixation indices** | | | | |
| $F = 0.18$ | | $\theta = 0.07$ | | $f = 0.12$ |
| **$F_{ST}$ (θ) among pairs of populations** | | | | |
| | Ghana | Mozambique | Peru | Uganda |
| Ghana | 0 | | | |
| Mozambique | 0.049 | 0 | | |
| Peru | 0.046 | 0.041 | 0 | |
| Uganda | 0.08 | 0.09 | 0.1 | 0 |

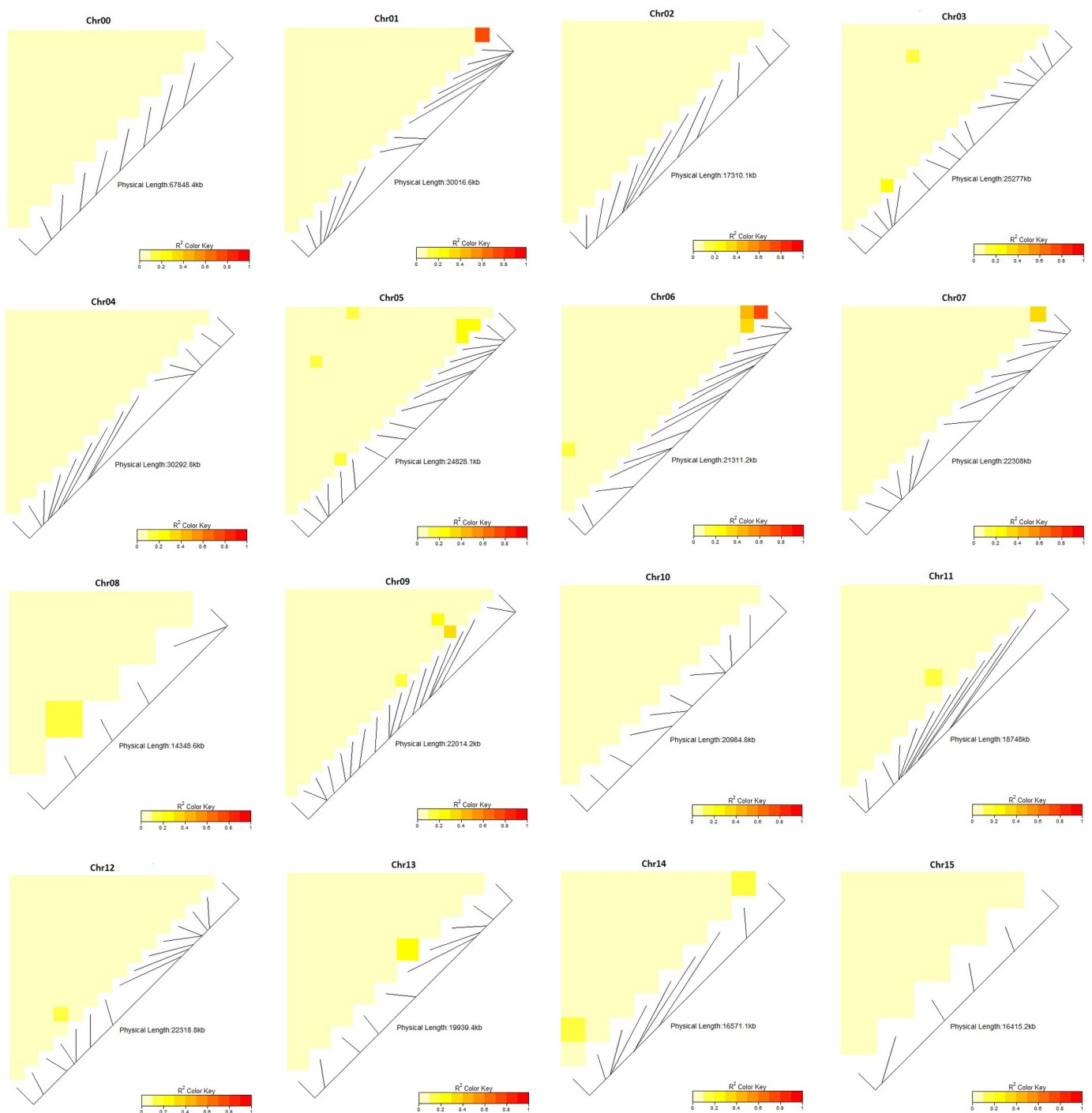

**Fig 3. Linkage disequilibrium among 205 markers used in analysis of population structure of 662 breeding parents of the International Potato Center's global breeding program.** The linkage disequilibrium was analyzed per chromosomes for the 15 base chromosomes of hexaploid sweetpotato.

evaluate the utility of the selected markers for QA/QC, we calculated Nei's coefficients of inbreeding using the 30 selected SNPs. The results showed the same trend as when using 205 SNPs, where, average $F_{IS} = 0.21$ across all populations. Parents from Uganda still had the

**Fig 4. Phylogenetic trees (Neighbor-Joining) comparing the clustering of the 662 International Potato Center (CIP)'s breeding parents using 205 highly informative SNP markers (left), 85-SNP intermediate marker set (center) and the 30-SNP selected QA/QC set markers (right).** Genotypes in Violet represent parents from the global support platform in Peru, genotypes in Green represent parents from the southern Africa support platform in Mozambique, genotypes in Orange represent parents from the east and central Africa support platform in Uganda, while genotypes in Blue represent parents from the west African support platform in Ghana. Trees were developed using DARwin 6.0.21.

highest inbreeding coefficient $F_{IS}$ = 0.35, followed by Mozambique with $F_{IS}$ = 0.27. Ghana, followed by Peru had the lowest coefficients of inbreeding at $F_{IS}$ = 0.08 and $F_{IS}$ = 0.13, respectively (data not shown).

## Estimated misclassification errors within the breeding process

Misclassification is used here to indicate mislabeling errors leading to distorted clustering of the same genotype *in vitro*, in the screen house and in the field. Cluster analysis indicated that 26 out of 94 genotypes evaluated did not cluster as expected among *in vitro*, screen house and field samples thereby indicating about 27.7% misclassification errors as the germplasm moved from *in vitro* to screenhouse and then to the field as can be seen from the Sankey diagram (**Fig 5, Top**). The Monte-Carlo tests comparing correlation among the separate distance matrices for *in vitro*, screenhouse and field indicated that more misclassification occurred when the genotypes moved from *in-vitro* (Nairobi, Kenya) to screenhouse when compared to movement from screenhouse to the field (both in Namulonge, Uganda). The correlation between distance matrices of *in vitro* and screenhouse was *r* = 0.84, between *in vitro* and field was *r* = 0.87 and between screenhouse and field *r* = 0.93. These indicated misclassification rates of 16%, 13% and 7% respectively (data not shown).

## Identifying diagnostic markers for routine quality assurance and control of breeding populations

Using 10,159 SNPs for QC as used above is 'rich' for routine QA/QC within most breeding programs. The desired low-cost, low-density QA/QC SNP set was therefore selected in this study based on the described criteria. This resulted in further filtration of the 205 SNP markers used for population structure of the parents above, down to 85 SNP markers (**S3 Data**), which could be used for 'general QC' in the sweetpotato breeding programs as proposed by [29]. However, for routine QC, this marker number is still probably too high for most breeding programs. Principal component analysis of the 85 markers did not show any specific grouping of markers, with PC1 and PC2 only explaining 9.1% of the variation (**S2 Fig**). Therefore, the final 30 SNP markers were selected based on genetic distance per chromosome. The set of 85 markers were not evenly distributed for all chromosomes and chromosome 15 had only one marker. To achieve the target of two markers per base chromosome, we selected one marker from the original set of 205 SNPs, based on genetic distance relative to the one marker present in the set of 85 SNPs, for chromosome 15. Comparing the population structure of the parents using 205,

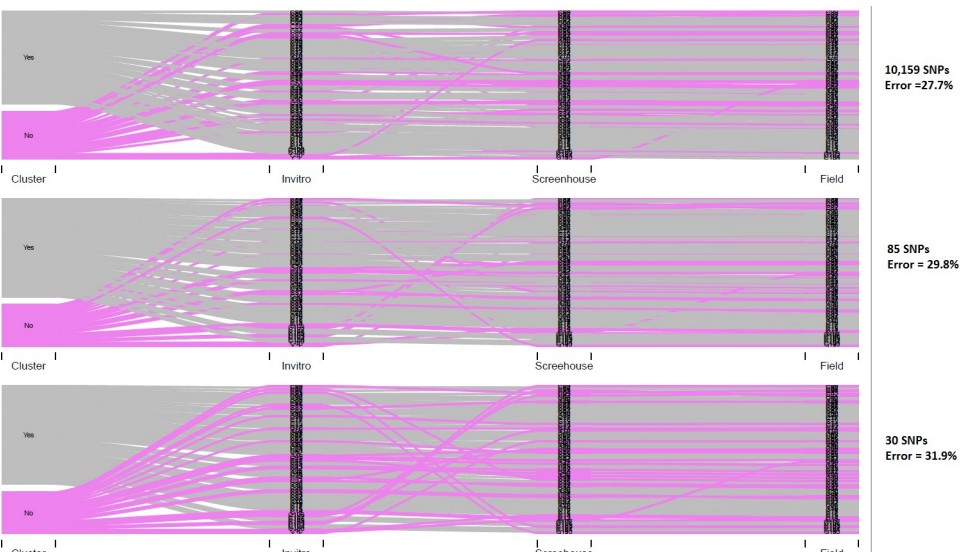

**Fig 5. Sankey diagrams showing mislabeling error as Mwanga diversity Panel (MDP) population moved from *in-vitro* to screen house to field, based on 10,159 SNPs, 85-SNP intermediate marker set and 30-SNP selected quality control (QC)-set.** The Pink color indicates those that did not cluster (with mislabeling errors) while the grey color indicates those that clustered as expected, implying no mislabeling errors.

85 and 30 SNP markers for parents indicated that the 30 markers kept the general structure of the populations, though the clustering was considerably different compared with the use of 205 SNPs (**Fig 4**). Comparing clustering trees indicated that the tree developed with 205 SNPs was 17.1% different from the tree with 30 SNPs when strict conditions were used. For validation of the selected marker set, we also compared the level of misclassification error identified in the breeding population progeny (MDP) using 10,159 SNPs, 85 SNPs and 30 SNPs (**Fig 5**). The misclassification error was calculated as the percentage of genotypes that did not cluster as expected in *in vitro*, screenhouse and field. Results show that 10,159 SNPs identified 27.7% misclassification, 85 SNPs identified 29.8% misclassification and 30 SNPs identified 31.9% misclassification. Tree comparison between 10,159 SNPs and 30 SNPs for MDP progeny indicated that they were 24.6% dissimilar when strict conditions were applied. Combined, these results suggest that the selected 30 SNPs could be used as a cost-effective rapid QA/QC set for sweetpotato in CIP's breeding populations, while trading-off the 4% difference in error detection between 10,159 SNPs and 30 SNPs, for a significantly reduced cost of QA/QC. The selected SNPs are listed in **Table 2**.

## Addition of trait specific markers to the selected quality control set

Previous studies had mapped quantitative trait loci (QTL) for yield and component traits [30, 31] as well as quality-related traits [32]. From these QTL mapping results, we selected six SNP markers that were associated with dry matter, starch, β-carotene, flesh color and total root yield. The markers labeled 'trait specific' are shown in **Table 2**. The first four traits were selected because they are correlated and important contributors to culinary traits that affect adoption of new varieties in sweetpotato. Dry matter and starch are positively correlated but are negatively correlated to both β-carotene and flesh color, and this negative correlation affects 'culinary quality'. Additionally, these traits are oligogenic hence results are repeatable within the QTL. Total storage root yield was selected as a primary trait and the selected marker

**Table 2. Details of the 36 SNPs selected for 'rapid QC' in sweetpotato.** AlleleID refers to the identity of the specific allele on the DArT platform, AlleleSeq refers to the flanking sequence of the SNP, Chr indicates the chromosome number, Pos indicates the position of a SNP on the specific chromosome, SNP is single nucleotide polymorphism.

| No | AlleleID | AlleleSeq | Chr | Pos | SNP |
|---|---|---|---|---|---|
| 1 | 7557698\|F\|0–64:T>A-64:T>A | TGCAGATAATAATACAAAACGTGATTTCTATTGTGCACCTAGAAGTGAGCAGAGTTGTCTGCCATAAGT | Chr01 | 30898063 | 64:T>A |
| 2 | 100736260\|F\|0–18:C>T-18:C>T | TGCAGTCAGCGACTCTCTCCAATGATATTCTTCTTCTGGAGCTGAGTGGAACTTCTTTCTTTGATTCTA | Chr01 | 881461 | 18:C>T |
| 3 | 7629110\|F\|0–28:G>T-28:G>T | TGCAGTCTTTGCTCTCAAAAGTTTCTTTGGAGTTCTCATATGAATTCTGAACATCACTAATTTGGATTG | Chr02 | 13184152 | 28:G>T |
| 4 | 7609930\|F\|0–10:T>G-10:T>G | TGCAGTCATCTGTTTGTCTGAAGCAATTAGCCTATGATCTTGTTGAGCTGCTGTTGTCATCTGCATTTC | Chr02 | 6247633 | 10:T>G |
| 5 | 7629039\|F\|0–39:A>T-39:A>T | TGCAGTACAGAAAACCAACCAGCAGAAGATAATTTTTATAATGAACAGCTCAGGAACCCAGTTGGCTAG | Chr03 | 24217089 | 39:A>T |
| 6 | 7561292\|F\|0–28:A>G-28:A>G | TGCAGTTGACTCATCCCAACCGACCTACACATTATCAAAACAATTACAGATCGGAAGAGCGGTTCAGCA | Chr03 | 3304180 | 28:A>G |
| 7 | 11826044\|F\|0–66:G>A-66:G>A | TGCAGTCCATATCAGAATGACAATTCTGTAGAGATTGCACAATCCTTTGGGTTTTCTTCTGCGTACGAT | Chr04 | 31341133 | 66:G>A |
| 8 | 7569592\|F\|0–50:G>A-50:G>A | TGCAGAAGATGGTGGTTGCGACAGAATGAAAGAATGGAGTAAGCAGAGAAGGCCATTACCCCTTCTGAT | Chr04 | 5305718 | 50:G>A |
| 9 | 7552489\|F\|0–18:T>G-18:T>G | TGCAGATAAAAGGTAAAATCAAACCACAAATCTAACTGTCCTCTACATTCCTTTCTATCAAATATTTGG | Chr05 | 24475925 | 18:T>G |
| 10 | 7562059\|F\|0–41:A>G-41:A>G | TGCAGATGAAATGAAATGAAAACTTTTAGTGCATATCATGTAAGCAATGTAATTGAAATCCACTAAGAG | Chr05 | 892499 | 41:A>G |
| 11 | 9847708\|F\|0–17:G>C-17:G>C | TGCAGAAAAACATACGCGGTGGATTGATGGTTCTCAAACAAATGGAAGATGCAGAAGTGAAACCTGACT | Chr06 | 19672316 | 17:G>C |
| 12 | 7558428\|F\|0–52:C>T-52:C>T | TGCAGCTACAACTTTGACAAGCTGGCATCTATTAGTTACGTTTTGTTCCCTTCATGTGGCACTCTTGAT | Chr06 | 4639839 | 52:C>T |
| 13 | 9845663\|F\|0–25:T>G-25:T>G | TGCAGTTTATCTAAGTAAGATGATATTCAGCGAGATGAAAACCCTAGGATGAGTGTGAAGGAATACAAG | Chr07 | 23485155 | 25:T>G |
| 14 | 7618077\|F\|0–38:G>A-38:G>A | TGCAGATCTTGAGCAGGTTGTAAATAAAGTGTGAGAGTGAATTAGTTACCACAATTCTTGTAAATTTAG | Chr07 | 5042144 | 38:G>A |
| 15 | 100588703\|F\|0–44:T>C-44:T>C | TGCAGGCAACTTTATTGAAATGTTGACTAAAATCTTGTTTTCTGTCAAGCTTCAACATAGACCTCATTG | Chr08 | 15171824 | 44:T>C |
| 16 | 100512185\|F\|0–24:A>G-24:A>G | TGCAGTATCCGAAATCCCTTTCCAAATGTTTGCTTATAAGCTGGTTGAGAAGGAGAAAAGTTTAGGGAA | Chr08 | 6218106 | 24:A>G |
| 17 | 7568783\|F\|0–21:T>A-21:T>A | TGCAGTGCATGCATGAGCCTCTGGCAACGTTGAGAAGTCACCCGCTTGCAGTTTCTCGGTCACGTCGGT | Chr09 | 22534529 | 21:T>A |
| 18 | 14313832\|F\|0–18:G>A-18:G>A | TGCAGATATAATGAAAAAGCACATAAAAAGTGACAAGAAATTATCAAATTAGGTACACTTGCTGCATCT | Chr09 | 520352 | 18:G>A |
| 19 | 7554048\|F\|0–9:G>A-9:G>A | TGCAGTATCGAAAGCAATGTCTTTGGTCTTCTTGTTAGGTTTCTCTTCCTTTTCCATTTCTATTTCACA | Chr10 | 4446069 | 9:G>A |
| 20 | 7574585\|F\|0–20:A>G-20:A>G | TGCAGAAACTCCCAAAGGAGATAGGAAATTTGCATCATCTAAGGTACATTGATTTACAGATCGGAAGAG | Chr10 | 6952705 | 20:A>G |
| 21 | 7619107\|F\|0–63:G>T-63:G>T | TGCAGTGACGATTCTTCCAATTAGCTCTTCTGCCCTTGAACAACAATCAAACATAACTAGCTTGCTGTT | Chr11 | 18928235 | 63:G>T |
| 22 | 7611165\|F\|0–24:G>T-24:G>T | TGCAGTCAATCAGATAGAACAATCGTTTAGTCTTTAGTTATGGTGATTGATAGGGGGAGTATACGATTA | Chr11 | 2783237 | 24:G>T |
| 23 | 7558251\|F\|0–66:C>A-66:C>A | TGCAGCCACGTGACACCAACAAACCCCTATTTTTCCGCCCAGTTTTGTTCTCACTTGGCGGGAAACCCC | Chr12 | 1719732 | 66:C>A |
| 24 | 7619930\|F\|0–17:C>A-17:C>A | TGCAGAGGATAAAAGTTCTGTACCCAAACAGGGGCTTTTTACAGATCGGAAGAGCGGTTCAGCAGGAAT | Chr12 | 24038510 | 17:C>A |
| 25 | 7562142\|F\|0–54:C>T-54:C>T | TGCAGATTGTGTAATCCCTTTAGAGTCAGCAACAGAGGCACTCTCGGTGATTCTCTTCTCATTATTATC | Chr13 | 22402544 | 54:C>T |
| 26 | 100589662\|F\|0–45:C>T-45:C>T | TGCAGTAATGATTTGGATATAGCACATACACATATAAATTATATACAATATAGTATTATTTTCAGCAAA | Chr13 | 7074575 | 45:C>T |

*(Continued)*

**Table 2.** (Continued)

| | | | Chr | Pos | SNP |
|---|---|---|---|---|---|
| 27 | 100619651|F|0–17:C>T-17:C>T | TGCAGTTGCTTAGCTTCCGCTACTTTGTTGGGTGGCCTTCTCTTTGCAGGTAATTTGAAGTACTAATCA | Chr14 | 17915206 | 17:C>T |
| 28 | 15728547|F|0–52:T>A-52:T>A | TGCAGTTTTATTGAAGCTGAAAGTTTGATCAGAGAGGGAGAGAGAGTTTGAGTGAGGAAAAGAATGAAG | Chr14 | 3121906 | 52:T>A |
| 29 | 7559173|F|0–7:T>C-7:T>C | TGCAGTATATGTATTATCAAATATGTGAAACGAGAATGATGACAGGTCAATCTAGAAGTGTAGCACATT | Chr15 | 11417254 | 7:T>C |
| 30 | 9845617|F|0–25:C>A-25:C>A | TGCAGTTCCTGCACTTCCAGTGAACCCCGATATATATGCTCTCCGCATATAACACTCAGCAATGAATTC | Chr15 | 8808402 | 25:C>A |

| | Trait-Specific Markers | | | | |
|---|---|---|---|---|---|
| | **Trait** | **Genetic Position** | **Chr** | **Pos** | **SNP** |
| 31 | Dry matter & Starch | 37.44 | Chr03 | 3185578 | C>T |
| 32 | β-Carotene & Flesh color | 36.14 | Chr03 | 2994719 | C>G |
| 33 | β-Carotene & Flesh color | 146.02 | Chr12 | 22131994 | G>A |
| 34 | Starch | 147.31 | Chr12 | 22197168 | T>A |
| 35 | Dry matter | 150.05 | Chr12 | 22369268 | A>T |
| 36 | Storage root yield | 4.19 | Chr15 | 452966 | A>C |

of a QTL was found to be a constitutive marker for this trait across several environments based on multi-environment testing of a full-sib population [30].

## Discussion

We obtained less than 10,000 bi-allelic SNP markers from genotyping our parental population. Stringent filtering resulted in an even smaller data set of 205 SNP markers. Similarly, only about 41,194 SNP markers were obtained from genotyping a segregating progeny, which were filtered down to 10,159 SNPs. This is in contrast with the high number of markers normally reported from GBS-based genotyping in diploid crops. The considerable reduction in highly informative markers can be associated with the difficulty in genotyping polyploids. With a mostly auto-hexaploid genome [19], sweetpotato presents allele dosage uncertainty due to ambiguous copy numbers of each allele. Additionally, the assumption of random inheritance of alleles may not hold true in this case due to uncharacterized consequences of whole genome duplication [33,34]. DArTSeq implements new protocols of sequencing complexity reduced representations [35] in combination with the next-generation sequencing methods [36,37]. Implementing genotyping-by-sequencing-like procedures, DArTSeq involves a two-restriction enzyme system composed of a 'rare-cutter' and a 'common-cutter', mainly *Pst1-Mse1*, to enhance uniform complexity reduction within the genome [38,39]. In such next-generation sequencing methods, depth of sequencing determines genotyping quality as low depth of coverage results in genotyping errors, misalignments and a lot of missing data which eventually cause biases in downstream population-genetic analyses [21, 40,41]. For instance, [42] showed that low sequencing depth resulted in SNPs that underestimated genomic heritability due to overestimation of inbreeding and underestimation of heterozygosity in ryegrass. We chose to use 'diploidized' data in the current analysis because the depth of coverage from most genotyping platforms is not adequate to reliably characterize heterozygous loci, such as those likely to be found in polyploids, for which deeper sequencing is required [43]. Furthermore, analyses comparing genotypic data from DArTSeq and those from a deep sequencing optimization platform for sweetpotato called GBSpoly [44] have confirmed that highly informative 'diploidized' DArTseq data performed just as well as high confidence data with dosage in genomic predictions of sweetpotato for traits with simple trait architecture [20]. This therefore implies

that such markers are adequate to carry out routine activities such as population structure and diversity that do not necessarily depend on details of allele dosage and haplotyping.

Population structure as well as allele diversity analyses in the current study indicated that parental genotypes from Uganda were the more distinct and inbred. This observation can be associated with the high sweetpotato virus disease (SPVD) pressure around the lake region of eastern Africa and a general lack of germplasm with high levels of resistance to SPVD necessitating the use of the same lines frequently as parents in the Ugandan breeding program [45–47]. SPVD is the most important virus complex in SSA and its effects are most pronounced in east Africa, causing yield loses of about 56–98% in farmers' fields [48,49]. [50] noted that SPVD prevalence in east Africa resulted in the failure of nearly all orange-fleshed varieties introduced into this region. SPVD is caused by a synergistic and complex infection by sweetpotato feathery mottle virus and sweetpotato chlorotic stunt virus, transmitted by aphids and white flies, respectively [51]. [52] indicated that different regions have different strains of the individual virus, and that east Africa has distinct strains. Studies have also showed that sweetpotato chlorotic stunt virus strains are more related in east and southern Africa and are distinct from those in the other regions of the world [53]. These results are supported by our current population structuring which shows that Uganda has some admixing with Mozambique, but very little admixing with either Peru or Ghana. These results have implications extending to other breeding decisions such as determining the effective population sizes especially for the Uganda breeding platform where migration of germplasm into the platform is restricted due to SPVD.

Different alleles are represented in different genetic backgrounds and our results show allele diversity between other support platforms with especially the Uganda population. Therefore, understanding population diversity especially of a global breeding program is important for breeding decisioning. Breeding programs are currently moving towards GAB. Repeatability of quantitative trait loci in different genetic backgrounds is one prerequisite for the success of GAB methods such as QTL mapping, genome-wide association mapping, and genomic selection [54,55]. In genome-wide association mapping, accounting for population structure avoids false positives and allows selection of causative variants, while accurate prediction of untested future genotypes in genomic selection is only possible when familial relatedness is accounted for, allowing for a reliable association between markers and QTL [56]. In the case of our global breeding population, the current information will be important when designing a genomic selection scheme to facilitate decisions such as prediction within or across sub-populations. Similarities and differences in genetic architecture of complex traits between populations can also be understood by studying the genetic correlation between the populations [55]. Our results indicate that the Ugandan sub-population was also the most distinct from the three others when $\theta$ values ($F_{ST}$) between pairs of populations was examined. This would imply that predictions may be carried out separately for the Ugandan populations in future GAB activities, while the predictions may be tested across the platforms in Peru, Mozambique and Ghana, given similar environmental conditions. Since GAB requires that markers be in LD with QTL, our results indicating very minimal LD among markers confirm that this marker density is not enough for making selection decisions [57,58]. Although the number of markers used in the current study are adequate for the purposes of the current objectives of population structuring, more dense markers along the genome will be required to reliably study the LD decay in sweetpotato. However, 'high density' has cost implications and hence the optimum number of markers required for routine GAB use will need to be reliably estimated through reducing within-haplotype density by selecting the minimum number of markers that can define common haplotypes [59].

In the current study, we used filtration and validation methods of DArTSeq developed markers to select a marker set of 30 SNPs that can be used for QA/QC purposes in sweetpotato. Our selection of informative markers included considerations for depth of coverage, missingness, chromosome position, genetic distances, validation for repeatability in progeny and inclusion of trait-specific markers to result in a total of 36 SNPs. Development of SNP sets for QA/QC has been done in several crops. Extensive tests were carried out to develop a SNP set for 'broad' and 'rapid' QC in maize [29]. In their study, they showed that marker coverage between 2 and 15, markers with less than 20% missing values, including markers with known chromosome positions, markers with less than 6% heterogeneity, inclusion of trait specific markers, and selection of markers from groups based on average group distance gave the best marker set towards developing a 'rapid' QC set, using DArTSeq markers. Prior to this, [1], used about 1,597 SNP markers from the KASPar and GoldenGate platforms to select highly informative markers for low-cost QC genotyping in maize. They recommended a set of 50–100 SNPs for routine QC after finding about 29% heterogeneity in inbred lines. In rice, [2] recommended a subset of 24–36 SNP markers filtered from DArTSeq developed markers for genetic purity analyses. In sweetpotato, QA/QC problems in breeding operations have been shown in the current manuscript by monitoring the rate of misclassification as materials moved through different stages of breeding trialing. Our results show about 30% misclassification issues in one breeding population. Routine application of QA/QC in sweetpotato breeding trials will therefore improve precision and breeding efficiency through use of new methods like forward breeding and genomic selection currently being adopted by CGIAR programs. These new breeding strategies are aimed towards increasing the rate of genetic gains from breeding to address issues related with population increase and climate change. Therefore, QA/QC of breeding processes will improve the likelihood of success.

Since the real impact from breeding can only be measured by the improvements observed in farmers' fields, controlling and assuring the quality of finished varieties is also important to breeding programs. Issues with QA/QC of released varieties have been reported in sweetpotato and this is exacerbated because the extent of adoption of new varieties cannot be determined accurately especially with informal seed systems where genetic integrity is seldom considered [60]. In Ethiopia, [18] used 17,220 DArTSeq developed markers to establish that about 20% of farmers confused local varieties for improved varieties and vice versa, and that farmers assigned different local names to the same variety or vice versa. Their study confirmed that data from survey studies [61,62] were mostly unreliable. Despite this important revelation, high density genotyping at 17,220 markers is not amenable for widespread routine use, therefore leaving household surveys as the predominant way of carrying out adoption studies. The currently developed marker set will therefore be useful in also addressing adoption-related needs in sweetpotato.

Towards deployment in breeding programs, the identified markers will be converted into competitive allele-specific PCR (KASP) system, for routine use of the target SNPs for QA/QC. However, to achieve increased genetic gains in the sweetpotato breeding programs, QA/QC will need to be combined with other approaches of optimizing breeding schemes.

## Supporting information

**S1 Data. The original DArTSeq data generated from 662 parents of the global breeding program of the international potato center.**
(CSV)

**S2 Data. DArTseq single nucleotide polymorphism (SNP) markers (raw data) from the segregating progeny used in quality control of breeding operations.**
(XLSX)

**S3 Data. Separate excel sheets showing the 205 stringently filtered and highly polymorphic SNPs (parents-205), 85-SNP intermediate with highlighted 30-SNP selected QA/QC set based on parents (Parents-85&Selected QC Set), and 85-SNP intermediate marker set with highlighted selected QA/QC set based on the progeny of the Mwanga Diversity Panel (MDP-85&Selected QC Set).**
(XLSX)

**S1 Fig. Two-dimensional figure from multidimensional scaling of the 662 International Potato center (CIP)'s global sweetpotato breeding parents as observed using 205 highly informative SNP markers.**
(PDF)

**S2 Fig. Principle component analysis (PCA) carried out on 85-SNP intermediate marker set to check for possible groupings that could aid the selection of a 30-SNP quality control marker set.**
(PDF)

**S1 Table. List of 662 parental genotypes from the International Potato Center (CIP)'s global breeding program including breeding support platform of origin.**
(XLSX)

**S2 Table. Genotype list and family assignments of the segregating progeny used in quality control of breeding operations.**
(XLSX)

**S1 File. The metadata that is associated with columns related to genotype quality parameters as provided by the DArTSeq platform.**
(XLSX)

## Acknowledgments

The work was carried out as part of the Consultative Group on International Agricultural Research (CGIAR)-Research Program on Roots, Tubers and Bananas (RTB). Sincere thanks to the Integrated Genotyping Service and Support (IGSS) platform for genotyping the populations. Additionally, the authors acknowledge all technical breeding teams in Ghana, Mozambique, Peru and Uganda. This manuscript has been released as a pre-print at **doi:** http://biorxiv.org/cgi/content/short/826792v1, (Gemenet et al.).

## Author Contributions

**Conceptualization:** Dorcus C. Gemenet, Hugo Campos.

**Data curation:** Dorcus C. Gemenet.

**Formal analysis:** Dorcus C. Gemenet.

**Funding acquisition:** Dorcus C. Gemenet, G. Craig Yencho, Simon Heck, Hugo Campos.

**Investigation:** Dorcus C. Gemenet, Mercy N. Kitavi, Maria David, Dorcah Ndege.

**Methodology:** Dorcus C. Gemenet, Mercy N. Kitavi, Maria David, Dorcah Ndege.

**Project administration:** Dorcus C. Gemenet, G. Craig Yencho.

**Resources:** Reuben T. Ssali, Jolien Swanckaert, Godwill Makunde, Wolfgang Gruneberg, Edward Carey, Robert O. Mwanga, Maria I. Andrade.

**Supervision:** Dorcus C. Gemenet, G. Craig Yencho, Hugo Campos.

**Visualization:** Dorcus C. Gemenet.

**Writing – original draft:** Dorcus C. Gemenet.

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
