## [Decision Letter · Decision Letter 0]

20 Jan 2020

PONE-D-19-31393

Development of diagnostic SNP markers for quality assurance and control in sweetpotato [Ipomoea batatas (L.) Lam.] breeding programs

PLOS ONE

Dear Dr. Gemenet,

Thank you for submitting your manuscript to PLOS ONE. After careful consideration, we feel that it has merit but does not fully meet PLOS ONE’s publication criteria as it currently stands. Therefore, we invite you to submit a revised version of the manuscript that addresses the points raised during the review process.

We would appreciate receiving your revised manuscript by Mar 05 2020 11:59PM. To enhance the reproducibility of your results, we recommend that if applicable you deposit your laboratory protocols in protocols.io, where a protocol can be assigned its own identifier (DOI) such that it can be cited independently in the future. For instructions see: http://journals.plos.org/plosone/s/submission-guidelines#loc-laboratory-protocols

We look forward to receiving your revised manuscript.

Kind regards,

Ajay Kumar

Academic Editor

PLOS ONE

Journal Requirements:

2. We noted in your submission details that a portion of your manuscript may have been presented or published elsewhere: The  results of 10,159 SNPs used to compare the  misclassification error rates identified in the current study using 85 SNPs and 30 SNPs for the MDP population (Fig 5 Top) had previously been used in a different manuscript for different objectives. The manuscript which is under review is available as a preprint and cited in the reference. It is also provided as related work with this submission. We do not see this as dual publication because in this manuscript, the previous data is only used to check if the selected QC SNP set is adequate for use in quality management of trials, whereas in the previous manuscript, it was used to identify misclassifications rates in a breeding population of interest.

The work was carried out as part of the Consultative Group on International Agricultural Research (CGIAR)-Research Program on Roots, Tubers and Bananas (RTB) which is supported by CGIAR Fund Donors (http://www.cgiar.org/about-us/our-funders/).

Genotyping for this work was funded by the SUSTAIN project awarded to the International Potato Center (SH) by Department for International Development (DFID) through grant Number (1198-DFID). Most co-authors were supported by a Bill & Melinda Gates Foundation (BMGF) grant (Grant number OPP1052983) awarded to North Carolina State University (GCY). The funders had no role in study design, data collection and analysis, decision to publish, or preparation of the manuscript.

Reviewers' comments:

Reviewer's Responses to Questions

**Comments to the Author**

1. Is the manuscript technically sound, and do the data support the conclusions?

Reviewer #1: No

Reviewer #2: Yes

Reviewer #3: Yes

2. Has the statistical analysis been performed appropriately and rigorously? 

Reviewer #1: I Don't Know

Reviewer #2: Yes

Reviewer #3: Yes

3. Have the authors made all data underlying the findings in their manuscript fully available?

Reviewer #1: No

Reviewer #2: Yes

Reviewer #3: Yes

4. Is the manuscript presented in an intelligible fashion and written in standard English?

Reviewer #1: No

Reviewer #2: Yes

Reviewer #3: Yes

5. Review Comments to the Author

Reviewer #1: I think the paper “Development of diagnostic SNP markers for quality assurance and control in sweetpotato [Ipomoea batatas (L.) Lam.] Breeding programs” is a good paper. However, I do not wish to print to PLOS ONE for the following reasons.

The development of the SNP marker, the subject of the study, is very difficult to apply in the hexaploid sweet potato. In addition, the scientists submitted the data with a lot of effort, but they did not present the phenotyping results decisively, so it could not be applied in a suitable breeding program.

I do not think there is any concern about double publishing, research ethics or publishing ethics.

Reviewer #2: Dear,

The authors developed a low-cost QA/QC SNP marker set for sweetpotato breeding programs. This marker set was selected by considering the depth of coverage, missingness, chromosome position, genetic distances and etc. Using the developed marker set, they evaluated genetic diversity in sweetpotato materials derived from Peru, Uganda, Mozambique, and Ghana, and the results provided new insights into genetic population structure in those materials.

I think this manuscript should provide informative data for QA/QC assessment, genetic studies and breeding programs in sweetpotato. However, I have one minor suggestion.

Page 8, Line numbers 178-179

I think the authors should describe more about the results for Fig1. Fig 1 shows the information of call rate, frequency of homozygous or heterozygous allele and PIC etc. How about adding sentences about differences and/or characteristics about these points between unfiltered and filtered SNP data?

Reviewer #3: The manuscript describes the developing of a new set of 36 molecular markers (SNPs), that is useful for genetic studies in sweetpotato breeding programs. Overall, the study is well-written and well-designed. The main strength of the present manuscript is the use of a diverse germplasm representing global breeding programs. Methods and conclusions are sounding, and results represent valid contributions to the sweetpotato community. As follow, I have two main considerations, that I believe could be used to improve the paper.

(i) Please, check all figures. They were presented in poor resolutions hindering the interpretation.

(ii) The authors should indicate how this new set of markers would be inserted into the scope of a breeding programs. More specifically, which genotyping platform do you propose to access such polymorphic regions in the breeding routine? For example, are you intend to include these makers in a pre-designed chip array or use rt-PCR derivative methods? In short, the manuscript lacks a discussion on that.

I have other points in reference to some lines throughout the text:

Figure 1- Please, cite in the Material and Methods how did you compute such quality attributes. Also, the different scale presented for the original vs. filtered plots are making all comparisons harder.

Line 133: Please, indicate how did you compute the PIC values using SNP data.

Line 174: Redundant information. Filtering criteria were previously described in the Material and Methods.

Line 208: Have you tried to estimate these same population parameters, but using the selected set of molecular markers?

Line 240: Redundant information. Filtering criteria were previously described in the Material and Methods.

Line 218: This information is confusing. First, I would suggest plotting all LD results in a LD decay format (r2 vs. base pair distance). Also, assuming that the current data set is structured in sub populations (as evidenced in the previous analyses), it could me more appropriate to present LD plots per population and minimize the population structure confounding.

Line 240: Did you check the functional effects of these SNPs ? It could be an additional biological criteria used for selecting potential markers.

Line 256: Please, define how the level of errors were computed and what is the meaningful for misclassifications.

Line 260: Based on Fig 4 and also on the percentage of dissimilarity reported, it is hard to affirm that the selected 30 SNPs are sufficient for genetic diversity studies. Visually, the tree presented in Fig 4c could not separate the genotypes in groups, as reported in the Fig. 4a. Please, clarify this point.

Line 296: This sentence needs more contextualization.

Line 311-318: This information is important. I would suggest including more details on that. For instance, in which context are the two methods similar? Diversity analysis?

6. PLOS authors have the option to publish the peer review history of their article (what does this mean?). If published, this will include your full peer review and any attached files.

Reviewer #1: Yes: Kyung-Min Kim

Reviewer #2: No

Reviewer #3: No

---

## [Author Response · Author response to Decision Letter 0]

4 Mar 2020

2. We noted in your submission details that a portion of your manuscript may have been presented or published elsewhere: The results of 10,159 SNPs used to compare the misclassification error rates identified in the current study using 85 SNPs and 30 SNPs for the MDP population (Fig 5 Top) had previously been used in a different manuscript for different objectives. The manuscript which is under review is available as a preprint and cited in the reference. It is also provided as related work with this submission. We do not see this as dual publication because in this manuscript, the previous data is only used to check if the selected QC SNP set is adequate for use in quality management of trials, whereas in the previous manuscript, it was used to identify misclassifications rates in a breeding population of interest.

Author Response: Thank you for this observation. These results have been previously published as a pre-print together with other objectives. The pre-print has not been accepted for publication by peer review. Since these results are more aligned with this manuscript, we have decided to report the results only in this manuscript and will remove it from the other objectives as previously presented in a preprint. The results will now be peer-reviewed only by PLOS ONE. The text has been updated to reflect this.

The work was carried out as part of the Consultative Group on International Agricultural Research (CGIAR)-Research Program on Roots, Tubers and Bananas (RTB) which is supported by CGIAR Fund Donors (http://www.cgiar.org/about-us/our-funders/).

Genotyping for this work was funded by the SUSTAIN project awarded to the International Potato Center (SH) by Department for International Development (DFID) through grant Number (1198-DFID). Most co-authors were supported by a Bill & Melinda Gates Foundation (BMGF) grant (Grant number OPP1052983) awarded to North Carolina State University (GCY). The funders had no role in study design, data collection and analysis, decision to publish, or preparation of the manuscript.

 Author Response: This is well noted and we apologize for the confusion. We wish to keep the funding statement as originally provided (highlighted in blue). The statement in the acknowledgement is only meant to acknowledge that all work is done within a crops research program (CRP) which is funded by several donors, but the CRP did not provide direct funding to the activities reported here in. We have deleted the last part of the statement that refers to donors of the CRP in the acknowledgement 

Reviewers' comments:

Reviewer's Responses to Questions

Comments to the Author

1. Is the manuscript technically sound, and do the data support the conclusions?

Reviewer #1: No

Reviewer #2: Yes

Reviewer #3: Yes

2. Has the statistical analysis been performed appropriately and rigorously? 

Reviewer #1: I Don't Know

Reviewer #2: Yes

Reviewer #3: Yes

3. Have the authors made all data underlying the findings in their manuscript fully available?

Reviewer #1: No

Reviewer #2: Yes

Reviewer #3: Yes

4. Is the manuscript presented in an intelligible fashion and written in standard English?

Reviewer #1: No

Reviewer #2: Yes

Reviewer #3: Yes

5. Review Comments to the Author

Reviewer #1: I think the paper “Development of diagnostic SNP markers for quality assurance and control in sweetpotato [Ipomoea batatas (L.) Lam.] Breeding programs” is a good paper. However, I do not wish to print to PLOS ONE for the following reasons.

The development of the SNP marker, the subject of the study, is very difficult to apply in the hexaploid sweet potato. In addition, the scientists submitted the data with a lot of effort, but they did not present the phenotyping results decisively, so it could not be applied in a suitable breeding program.

I do not think there is any concern about double publishing, research ethics or publishing ethics.

Author Response: We would like to thank reviewer 1 for time and effort to contribute to the betterment of our manuscript. However, since we have not done any correlation between phenotypic and genotypic data in the current manuscript, we could not fully address the concern of the reviewer as this is beyond the scope of the current manuscript.

Reviewer #2: Dear,

The authors developed a low-cost QA/QC SNP marker set for sweetpotato breeding programs. This marker set was selected by considering the depth of coverage, missingness, chromosome position, genetic distances and etc. Using the developed marker set, they evaluated genetic diversity in sweetpotato materials derived from Peru, Uganda, Mozambique, and Ghana, and the results provided new insights into genetic population structure in those materials.

I think this manuscript should provide informative data for QA/QC assessment, genetic studies and breeding programs in sweetpotato. However, I have one minor suggestion.

Author Response: We are grateful to the reviewer for their time to improve our manuscript and for finding value in our manuscript.

Page 8, Line numbers 178-179

I think the authors should describe more about the results for Fig1. Fig 1 shows the information of call rate, frequency of homozygous or heterozygous allele and PIC etc. How about adding sentences about differences and/or characteristics about these points between unfiltered and filtered SNP data?

Author Response: This suggestion has been considered and additional description of Fig 1 added to text as can be accessed in the copy with track changes.

Reviewer #3: The manuscript describes the developing of a new set of 36 molecular markers (SNPs), that is useful for genetic studies in sweetpotato breeding programs. Overall, the study is well-written and well-designed. The main strength of the present manuscript is the use of a diverse germplasm representing global breeding programs. Methods and conclusions are sounding, and results represent valid contributions to the sweetpotato community. As follow, I have two main considerations, that I believe could be used to improve the paper.

Author Response: we thank the reviewer for finding time to contribute to the improvement of our manuscript and finding value. We address their considerations as below.

(i) Please, check all figures. They were presented in poor resolutions hindering the interpretation.

Author Response: We have addressed this by uploading the figures to PACE as advised by the editorial office. https://pacev2.apexcovantage.com/. PACE helps ensure that figures meet PLOS requirements.

(ii) The authors should indicate how this new set of markers would be inserted into the scope of a breeding programs. More specifically, which genotyping platform do you propose to access such polymorphic regions in the breeding routine? For example, are you intend to include these makers in a pre-designed chip array or use rt-PCR derivative methods? In short, the manuscript lacks a discussion on that.

Author Response: This is a very important point. We have now clarified in text our plans to deploy these into breeding programs, by converting them into KASP system. This can be seen in text with track changes.

I have other points in reference to some lines throughout the text:

Figure 1- Please, cite in the Material and Methods how did you compute such quality attributes. Also, the different scale presented for the original vs. filtered plots are making all comparisons harder.

Author Response: We have described these in text and also attached the metadata that describes all data columns as provided by the DArTSeq platform, as supplementary

Line 133: Please, indicate how did you compute the PIC values using SNP data.

Author Response: This has been clarified in text as being one of the outputs from the DArTSeq proprietary platform

Line 174: Redundant information. Filtering criteria were previously described in the Material and Methods.

Author Response: The repeated information has been deleted from M&M.

Line 208: Have you tried to estimate these same population parameters but using the selected set of molecular markers?

Author Response: This is a good suggestion. We have calculated Nei’s coefficients of inbreeding also using the 30 selected markers and observed the same trends in the population. This has been updated in the text.

Line 240: Redundant information. Filtering criteria were previously described in the Material and Methods.

Line 218: This information is confusing. First, I would suggest plotting all LD results in a LD decay format (r2 vs. base pair distance). Also, assuming that the current data set is structured in sub populations (as evidenced in the previous analyses), it could me more appropriate to present LD plots per population and minimize the population structure confounding.

Author Response: We have rephrased this in the text to indicate that the LD was carried out to check the distribution of markers genome-wide or within a chromosome. We did not calculate LD per population as the objective of this manuscript is not to determine LD decay in sweetpotato, which would require higher marker densities and other considerations of dosage information. This information, which would be very important if we had an objective of genome-wide association study is still being considered among different sweetpotato groups (most co-authors here) to resolve issues around dosage and haplotype information, as part of genomic tool development. For the current manuscript, the aim is only to select representative markers for population structure/diversity, which do not necessarily suffer from lack of allele dosage information and linkage phases. This is also acknowledged and discussed in the discussion section. We hope this clarification addresses the reviewer concern.

Line 240: Did you check the functional effects of these SNPs ? It could be an additional biological criteria used for selecting potential markers.

Author Response: This is a very good suggestion. For the current diploidized SNPs, we did not check for any functional effects, given that function in hexaploid sweetpotato will also depend on issues such as dosage, linkage phase and present haplotypes. However, we added six trait-specific markers to the list which we selected from high effect QTL for the traits in a biparental population, using dosage information and a well phased linkage map. However, even for these, we are still resolving issues with dosage and haplotypes before they can be converted into a marker system for routine use in trait selection. Including them in this QA/QC set will generate additional information to support these decisions.

Line 256: Please, define how the level of errors were computed and what is the meaningful for misclassifications.

Author Response: We have included in text that misclassification means mislabeling errors, as suggested by the reviewer. We have also clarified that the error was computed by taking the percentage of genotypes that did not cluster as expected in in vitro, screenhouse and field. The clustering based on different marker sets was also correlated using a tree comparison program.

Line 260: Based on Fig 4 and also on the percentage of dissimilarity reported, it is hard to affirm that the selected 30 SNPs are sufficient for genetic diversity studies. Visually, the tree presented in Fig 4c could not separate the genotypes in groups, as reported in the Fig. 4a. Please, clarify this point.

Author Response: This is a very good point, especially, to do with resource allocation in a breeding program. Moving from 10k SNPs to 30, while losing about 24% precision is a trade-of most breeding programs would take, especially given that 10k is only 4% better in identifying misclassification. This is as opposed to not doing QA/QC activities at all due to costs.

Line 296: This sentence needs more contextualization.

Author Response: This is noted and rephrased in text.

Line 311-318: This information is important. I would suggest including more details on that. For instance, in which context are the two methods similar? Diversity analysis?

Author Response: This has been rephrased in text for clarity and the manuscript which was under preparation is now available as a preprint and cited in text as well as referenced.

6. PLOS authors have the option to publish the peer review history of their article (what does this mean?). If published, this will include your full peer review and any attached files.

Do you want your identity to be public for this peer review? For information about this choice, including consent withdrawal, please see our Privacy Policy.

Reviewer #1: Yes: Kyung-Min Kim

Reviewer #2: No

Reviewer #3: No

Thank you for the suggestion. This has been done.

---

## [Decision Letter · Decision Letter 1]

9 Apr 2020

Development of diagnostic SNP markers for quality assurance and control in sweetpotato [Ipomoea batatas (L.) Lam.] breeding programs

PONE-D-19-31393R1

Dear Dr. Gemenet,

We are pleased to inform you that your manuscript has been judged scientifically suitable for publication and will be formally accepted for publication once it complies with all outstanding technical requirements.

With kind regards,

Ajay Kumar

Academic Editor

PLOS ONE

Additional Editor Comments (optional):

Reviewers' comments:

Reviewer's Responses to Questions

**Comments to the Author**

1. If the authors have adequately addressed your comments raised in a previous round of review and you feel that this manuscript is now acceptable for publication, you may indicate that here to bypass the “Comments to the Author” section, enter your conflict of interest statement in the “Confidential to Editor” section, and submit your "Accept" recommendation.

Reviewer #2: (No Response)

Reviewer #3: All comments have been addressed

2. Is the manuscript technically sound, and do the data support the conclusions?

Reviewer #2: (No Response)

Reviewer #3: Yes

3. Has the statistical analysis been performed appropriately and rigorously? 

Reviewer #2: (No Response)

Reviewer #3: Yes

4. Have the authors made all data underlying the findings in their manuscript fully available?

Reviewer #2: (No Response)

Reviewer #3: Yes

5. Is the manuscript presented in an intelligible fashion and written in standard English?

Reviewer #2: (No Response)

Reviewer #3: Yes

6. Review Comments to the Author

Reviewer #2: (No Response)

Reviewer #3: Dear authors,

I appreciate the opportunity to re-review the manuscript: “Development of diagnostic SNP markers for quality assurance and control in sweetpotato [Ipomoea batatas (L.) Lam.] breeding programs”. This is my second review of the article. The authors have made the recommended changes in this new version. I have no objection to the methods or results that were presented. Finally, I appreciate the kindness of all authors to clarify my comments/questions.

Best regards,

Felipe Ferrão

7. PLOS authors have the option to publish the peer review history of their article (what does this mean?). If published, this will include your full peer review and any attached files.

Reviewer #2: No

Reviewer #3: Yes: Luis Felipe Ventorim Ferrão

---

## [Editor Report · Acceptance letter]

10 Apr 2020

PONE-D-19-31393R1 

 Development of diagnostic SNP markers for quality assurance and control in sweetpotato [*Ipomoea batatas* (L.) Lam.] breeding programs 

Dear Dr. Gemenet:

I am pleased to inform you that your manuscript has been deemed suitable for publication in PLOS ONE. Congratulations! Your manuscript is now with our production department. 

With kind regards,

on behalf of

Dr. Ajay Kumar 

Academic Editor

PLOS ONE